# Function of histone H2B monoubiquitination in transcriptional regulation of auxin biosynthesis in Arabidopsis

Li Zhang [1], Pan Luo[1,2], Jie Bai[1], Lei Wu [1], Dong-Wei Di[1,3], Hai-Qing Liu[1], Jing-Jing Li[1], Ya-Li Liu[1], Allah Jurio Khaskheli[1], Chang-Ming Zhao [1,4 ✉] & Guang-Qin Guo [1 ✉]

The auxin IAA is a vital plant hormone in controlling growth and development, but our knowledge about its complicated biosynthetic pathways and molecular regulation are still limited and fragmentary. *cytokinin induced root waving* 2 (*ckrw2*) was isolated as one of the auxin-deficient mutants in a large-scale forward genetic screen aiming to find more genes functioning in auxin homeostasis and/or its regulation. Here we show that *CKRW2* is identical to *Histone Monoubiquitination 1 (HUB1)*, a gene encoding an E3 ligase required for histone H2B monoubiquitination (H2Bub1) in Arabidopsis. In addition to pleiotropic defects in growth and development, loss of CKRW2/HUB1 function also led to typical auxin-deficient phenotypes in roots, which was associated with significantly lower expression levels of several functional auxin synthetic genes, namely *TRP2/TSB1*, *WEI7/ASB1*, *YUC7* and *AMI1*. Corresponding defects in H2Bub1 were detected in the coding regions of these genes by chromatin immunoprecipitation (ChIP) analysis, indicating the involvement of H2Bub1 in regulating auxin biosynthesis. Importantly, application of exogenous cytokinin (CK) could stimulate *CKRW2/HUB1* expression, providing an epigenetic avenue for CK to regulate the auxin homeostasis. Our results reveal a previously unknown mechanism for regulating auxin bio-synthesis via HUB1/2-mediated H2Bub1 at the chromatin level.

[1] Institute of Cell Biology and MOE Key Laboratory of Cell Activities and Stress Adaptations, School of Life Sciences, Lanzhou University, Lanzhou, Gansu, P.R. China. [2] College of Life Science and Technology, Gansu Agricultural University, Lanzhou, Gansu, P.R. China. [3] State Key Laboratory of Soil and Sustainable Agriculture, Institute of Soil Science, Chinese Academy of Sciences, Nanjing, P.R. China. [4] State Key Laboratory of Grassland Agro-Ecosystems, School of Life Sciences, Lanzhou University, Lanzhou, P.R. China. ✉email: zhaochm@lzu.edu.cn; gqguo@lzu.edu.cn

Auxin is one of the most important plant hormones regulating plant growth and development, such as cell division, cell differentiation, apical dominance, flowering, senescence, and tropism[1–4]. Plants maintain auxin homeostasis by regulating its synthesis, metabolism, and polar transport[5,6].

Plants are believed to have multiple and highly interconnected pathways for auxin biosynthesis, including several tryptophan (Trp)-dependent (TD) and -independent (TI) pathways[7]. In these pathways, Trp is synthesized from chorismate via six critical linear steps in the chloroplast[8]. The *WEI2/ASA1* and *WEI7/ASB1* genes encode the α- and β-subunit, respectively, of the anthranilate synthetase complex, which catalyzes the rate-limiting step in the conversion of chorismate to anthranilate. Under the catalysis of PAT1 and PAI, anthranilate is converted to CdRP[9,10]. Subsequently, the indole glycerol phosphate synthetase catalyzes the conversion of CdRP to IGP, which is the branch point of TI and TD pathways. IGP can form Trp through Trp synthetase complex, which is composed of Trp synthase α (TSA1) and β (TSB1 and TSB2)[11,12]. According to the intermediate metabolites from Trp, the TD pathway can be divided into four branch pathways: the indole-3-pyruvic acid (IPyA), indole-3-acetamide (IAM), indole-3-acetaldoxime (IAOx), and tryptamine (TAM) pathways[10,13,14]. So far, only the two-step IPyA pathway has been completely elucidated at both the biochemical and the genetic level, producing IAA from Trp via IPyA under the catalysis by tryptophan aminotransferases (TAA1/TARs) and YUCCA (YUC) flavin-dependent monooxygenases, and is likely the main pathway for auxin synthesis in Arabidopsis[15–22].

The synthesis of auxin in plants is subject to intricate regulations[1,6,23]. Nutritional signals such as glucose and nitrate induce the production of auxin by regulating the transcription of *YUC2/8/9* and *TAA1/TAR2*, respectively[24–27]. Environmental stress, such as aluminum, regulates the level of auxin by regulating the transcription of *TAA1*[28]. The plant hormone cytokinin (CK) regulates the level of auxin by regulating the transcription level of *YUC1/4/8* and *TAA1*[18,20,29–31]. These findings indicate that the transcriptional regulation of auxin synthase plays an important role in auxin homeostasis.

The epigenetic state of chromatin associated with histone modifications can profoundly influence gene expression in eukaryotes. Histone H2B monoubiquitination (H2Bub1) is a form of post-translational modification that is linked to active gene transcription[32,33]. In Arabidopsis, H2Bub1 normally occurs on K143 or K145[34] by the heterodimeric HISTONE MONO-UBIQUITINATION1/2 (HUB1/2) E3 ubiquitin ligase, a homolog of the budding yeast Bre1 protein[35,36]. For instance, HUB1/2-mediated H2Bub1 regulates the expression levels of *FLOWERING LOCUS C* (*FLC*) and some circadian clock genes such as *CCA1* and *TOC1* by stimulating the H3K4me3 modification on their chromatin[37–39]. In plant immune responses, H2Bub1 modulates the expression of the R gene *SNC1*[40]. H2Bub1 is reported to be associated with H3K4me3 at the GhDREB locus, which triggers more rapid responses to drought stress[41]. Such genome-wide regulation on gene expression makes HUB1/2 to be required for multiple developmental processes in plants, including seed dormancy[35], leaf, and root growth[36], flowering[37,42,43], photomorphogenesis, and circadian rhythms[38,44], defense and immune responses[40,45–47], and their loss of function mutation can produce a wide variety of defective phenotypes in growth and development.

Here, on characterizing a previously isolated auxin-deficient mutant[48] of *ckrw2*, we reveal that *CKRW2* is identical to *HUB1*, and HUB1/2-mediated H2Bub1 is positively associated with the transcription of several auxin synthetic genes for maintaining normal auxin homeostasis. By up-regulating *HUB1/2* gene expression, CK can use this kind of histone modification as one of its effective ways to regulate auxin homeostasis in plants.

## Results and discussion

***ckrw2* is an auxin-deficient mutant.** To uncover more genes functioning in auxin biosynthesis or homeostasis, we previously established an effective genetic screening protocol for isolating auxin-deficient mutants by using CK-induced root curling (ckrc) or root waving (ckrw) as a phenotypic marker, in which the *ckrw2* mutant was isolated as one of the so-called group III *ckrw* mutants[48]. When grown on the medium containing 0.01 μM trans-zeatin (tZ), *ckrw2* mutant displayed a root waving phenotype and had a significantly reduced endogenous IAA level[48]. In addition to a number of pleiotropic abnormalities in leaves, seeds, root hair, apical hook, cutin, petals, and flowering time (Supplementary Fig. 1 a–l), typical low-auxin phenotypes, such as the reduced root length, smaller meristematic zone, shorter mature epidermal cell length (Supplementary Fig. 1m–p) and weaker gravitropic response, were observed, which could be rescued by exogenous auxins[48] (Fig. 1a, b). In line with these, the mutant had weaker *Dr5:GUS/GFP* expression[49] or brighter *DII-VENUS*[50] fluorescence (Fig. 1c, d, Supplementary Fig. 2) in the transgenic root tips, indicating a lower auxin activity that was most likely caused by the endogenous auxin deficiency.

***CKRW2* gene encodes a functional E3 ubiquitin ligase for histone H2Bub1.** As *ckrw2* was isolated from a mutant pool generated by T-DNA tagging[51] we initially did Tail-polymerase chain reaction (PCR) amplification, finding a T-DNA flanking sequence located between AT5G25425-AT5G25430, which showed no genetic linkage to *ckrw2* mutation[48]. However, map-based cloning combined with whole-genome resequencing (WGRS) identified a G > A substitution in the coding region of *AT2G44950/HUB1*, altering the tryptophan (aa 91) codon (TGG) to a stop codon (TAG) (Fig. 2a) in this gene. Both genetic allelic analysis (Fig. 2b) and the full rescue of the defective *ckrw2* phenotypes (Fig. 2c; Supplementary Fig. 3) by the fused HUB1::YFP-HUB1 confirmed the At2g44950/HUB1 identity of *CKRW2* gene.

In Arabidopsis HUB1 and its paralog HUB2 are E3 ubiquitin ligases to act non-redundantly in a conserved heterotetrameric complex to catalyze H2Bub1 in chromatin, activating a variety of genes functioning in diverse biological processes of growth, development, stress, and immunity response[33,35–38,40,44–47,52–57], some of which were also observed in *ckrw2* mutant (Supplementary Figs. 1 and 3). *In planta*, CKRW2/HUB1 had a constitutive or wide expression pattern at the organ/tissue levels, as revealed by the *pCKRW2:GUS* reporter transgene expression (Fig. 2d–j)[35,37]. It was highly active in the meristematic and vascular tissues of the primary root, hypocotyl, stem, cotyledon, and leaves (Fig. 2d–j). Significantly, the expression of *CKRW2* at the apical root overlapped some of the auxin synthetic genes (Fig. 2e)[10,18,20,58]. As expected, the global defect in H2Bub1 was confirmed by western blot analysis using the anti-H2Bub1 antibody in *ckrw2* mutant (Supplementary Figs. 4 and 5)[37].

**CKRW2/HUB1 activates the transcription of *TSB1*, *WEI7*, *AMI1*, and *YUC7* through H2Bub1.** To investigate how *ckrw2* mutation affected auxin homeostasis, we measured the expression of a number of known auxin biosynthesis genes[2] by qRT-PCR (Fig. 3a and Supplementary Fig. 6), detecting significant reductions in the expression levels of *TRP2/TSB1*, *WEI7/ASB1*, *YUC7*, and *AMI1* (Fig. 3a), which functioning at distinct steps in the complex tryptophan/auxin biosynthetic pathways, either upstream of L-Trp biosynthesis (ASB1/WEI7 and TSB1/TRP2)[10,13], or downstream of it in the IPA pathway (YUC7) or the proposed

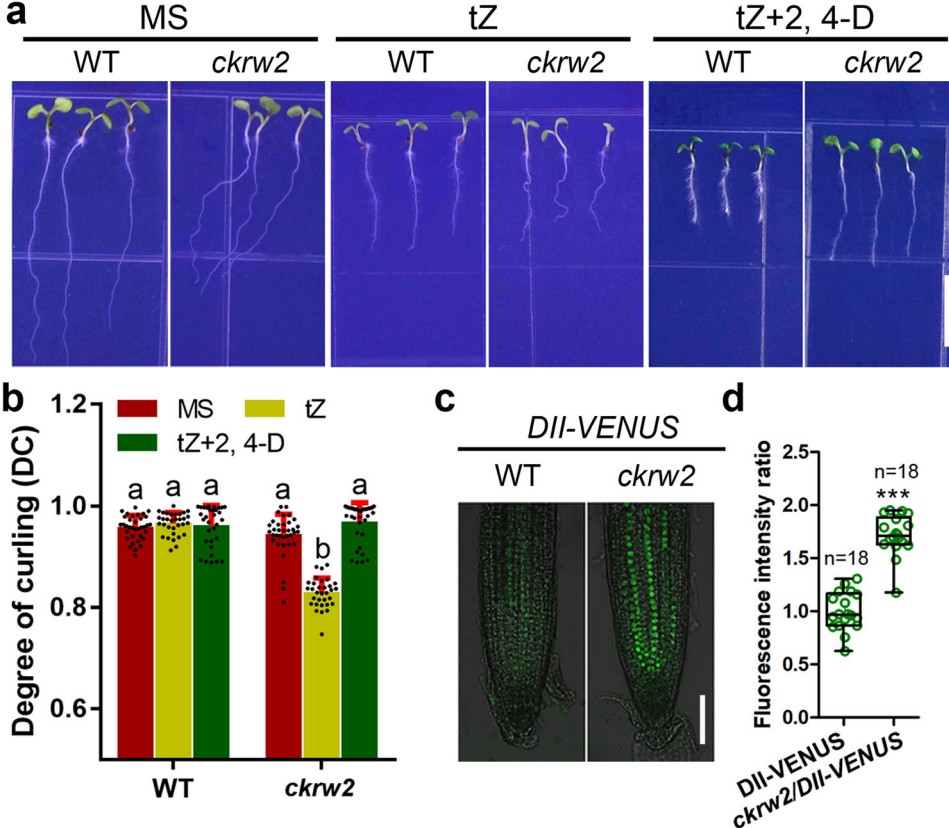

**Fig. 1 Auxin-deficient phenotypes in *ckrw2* mutant. a** Auxin-deficient phenotypes in *ckrw2* mutant, showing its waving roots after 7 days of vertical culture on MS with 0.01 μM tZ, which can be rescued by 0.01 μM 2, 4-D. Bar = 5 mm. **b** Results of statistical analysis on root curling/waving degrees (DC). Data are presented as mean ± SD ($n = 30–35$), three biologically independent experiments, the letters indicate a significant difference at $P < 0.05$ according to ANOVA followed by Tukey's multiple comparison tests. **c**, **d** Auxin activity in root tips revealed by the expressions of *DII-VENUS*. The seedlings grown on MS medium for 7 days were used for GUS staining and fluorescence intensity observation (approximately to the first 200 μm from the root tip). Each circle represents the measurement from an individual root. Boxplots span the first to the third quartiles of the data. Whiskers represent the minimum and maximum values. A line in the box represents the mean. "*n*" represents the number of roots used in this experiment. The student's *t* test, was used for statistical analyses. ***$P < 0.001$. Bars = 50 μm in (**d**).

IAM pathway (AMI1)[2,59–62]. Subsequent ChIP analysis detected a significantly lower amount of H2Bub1-associated DNA fragment in the coding but not 5′ upstream promoter or untranscribed regions of the four affected genes in *ckrw2* mutant (Fig. 3b, c), which is a prominent feature of histone H2Bub1 modification in affecting gene activity in the process of transcriptional elongation. These data demonstrate that *YUC7*, *TSB1*, *WEI7*, and *AMI1* in the auxin biosynthesis pathways are targeted by HUB1-mediated H2Bub1.

To clarify the functional roles of each of the four affected genes in HUB1/2-mediated regulation on auxin homeostasis, we did the mutant analysis. Among their loss of function mutants, *tsb1* and *wei7* had slightly more obvious auxin-deficient phenotypes of *ckrw2*-like curling/waving primary roots with a reduced length, but *yuc7* and *ami1* had not (Supplementary Fig. 8a–c), suggesting that *WEI7* and *TSB1* are the two major functional genes in H2Bub1-mediated regulation on auxin biosynthesis. These two genes, encoding the β-subunit of anthranilate synthetase (*WEI7*/ *ASB1*) complex and tryptophan synthase β (*TSB1*), respectively, are required for L-Trp biosynthesis[8], and their roles in auxin biosynthesis[10,63] and/or root waving[64] have been confirmed. Consisting with that, *ckrw2 tsb1* double mutant displayed very similar or the same phenotype of *tsb1* single mutant (Supplementary Fig. 8a–c). Moreover, like *wei7* and *tsb1*, L-TRP can rescue *ckrw2*, but not *ckrc1* in *Dr5:GUS* expression and plant growth analyses (Fig. 4c; Supplementary Fig. 9), indicating that

*CKRW2* affects auxin homeostasis through regulating *WEI7*/ *ASB1* and *TSB1* for L-Trp biosynthesis.

**The expression of CKRW2 are induced by CK.** The above results promoted us to study how the regulation of auxin biosynthesis by CK was related to H2Bub1. Some mechanisms have been revealed for CK-mediated regulation on auxin production, mostly via transcriptional factors[31,65,66]. The qRT-PCR results (Fig. 4a) and both of GUS staining to detect the *pCKRW2:GUS* expression (Supplementary Fig. 8d) and YFP fluorescence intensity to detect the *pHUB1::YFP-HUB1* expression (Supplementary Fig. 8e) showed that tZ treatment can stimulate *HUB1/ CKRW2* expression and increase the HUB1 protein level (Fig. 4b and Supplementary Fig. 10), leading to an increase of H2Bub1 activity. Consequently, the expression of *TSB1* and *WEI7* were significantly upregulated, which was not observed in *ckrw2* mutant (Fig. 4d), revealing that this upregulation depends on *CKRW2/HUB1* function.

In summary, our present studies reveal a mechanism at the chromatin level via H2Bub1 to control transcription of auxin biosynthesis genes. In this process, H2B proteins in the chromatin wrapped by the DNAs of auxin biosynthesis genes of *WEI7* and *TSB1* can be monoubiquitinated by HUB1/2 heterotetramer after the recruitment of UBC1/2[37], activating the transcriptional elongation of these genes[33]. Significantly, such an epigenetic

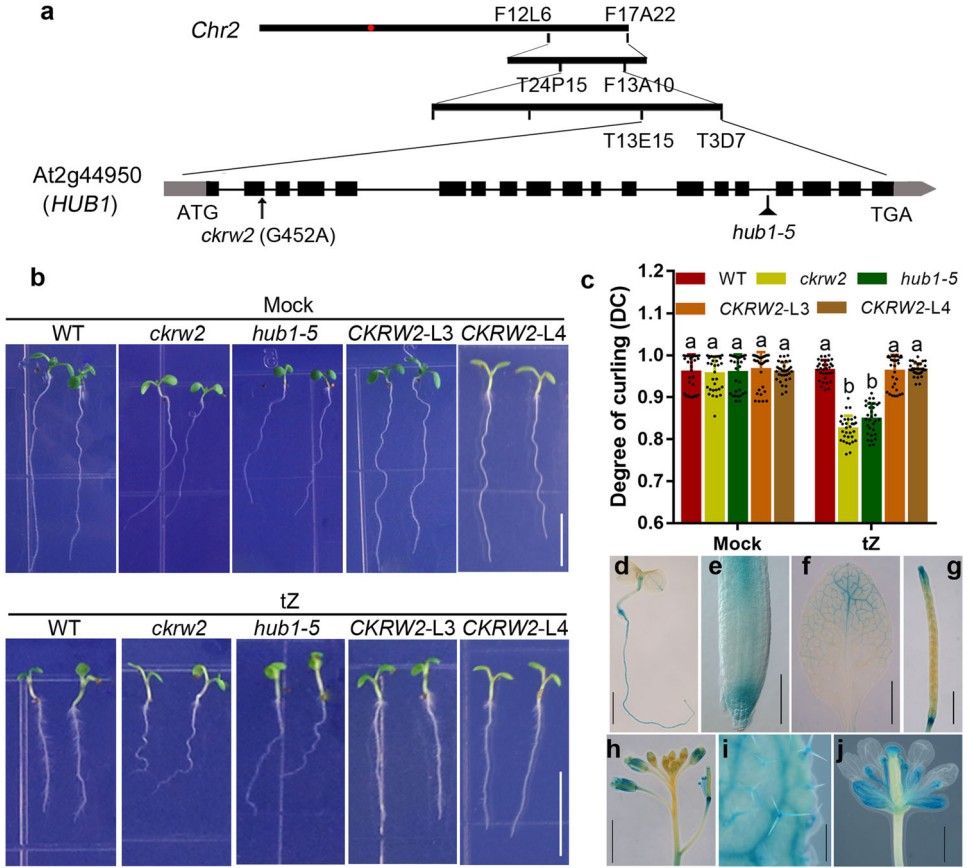

**Fig. 2 Gene cloning and the expression pattern of *CKRW2* gene. a** Map-based cloning, showing the position of the G452A point mutation causing premature termination in the *AT2G44950* gene coding region. **b, c** Similar phenotypes of *ckrw2* and *hub1-5* (**b**), and molecular complementation (**c**). Seedlings were grown on MS with or without 0.01 μM tZ. Bars = 5 mm. Data are presented as mean ± SD (*n* = 30–35), three independent experiments, the letters indicate a significant difference at *P* < 0.05, according to ANOVA followed by Tukey's multiple comparison tests. **d–j** GUS activity was detected in various organs at different developmental stages of *pCKRW2: GUS* transgenic plant. 4-day-old seedlings (**d–g**) or 4-week-old plants (**h–j**). Scale bars = 5 mm in (**d**) and (**j**); 100 μm in (**e**); 20 mm in (**f**) and (**h**); 10 mm in (**g**).

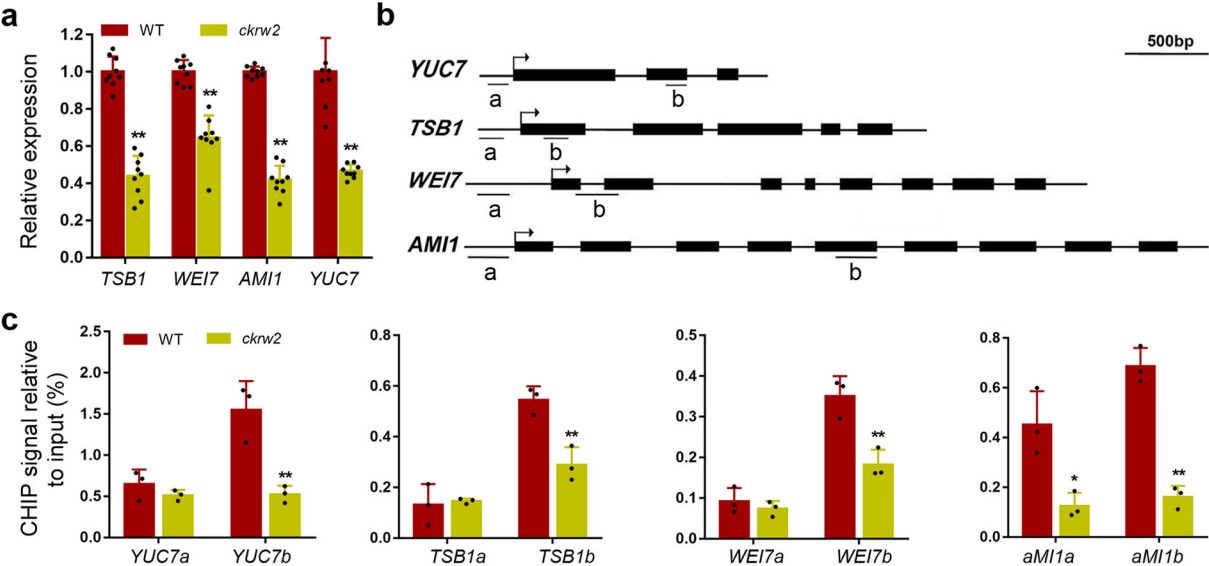

**Fig. 3 Reduced expression of auxin synthesis genes in *ckrw2* and analysis of H2Bub1 at these loci. a** Relative transcription levels of *TSB1*, *WEI7*, *AMI1*, and *YUC7* genes in roots. *ACTIN8* was used as an internal control. **b** Diagram representing the genomic structure and regions analyzed by ChIP assays, arrows indicate ATG start codon sites, and bars labeled "a" or "b" represent regions amplified by RT-qPCR in ChIP analysis. **c** H2Bub1 deposition at specific loci. *LCRa* and *FLCP4* were used as a negative and positive control, respectively (shown in Supplementary Fig. 7). Data are presented as mean ± SD, three independent biological experiments, the asterisk indicates a significant difference based on Student's *t* test with **\*\***P < 0.01, 0.01 < *\*P* < 0.05.

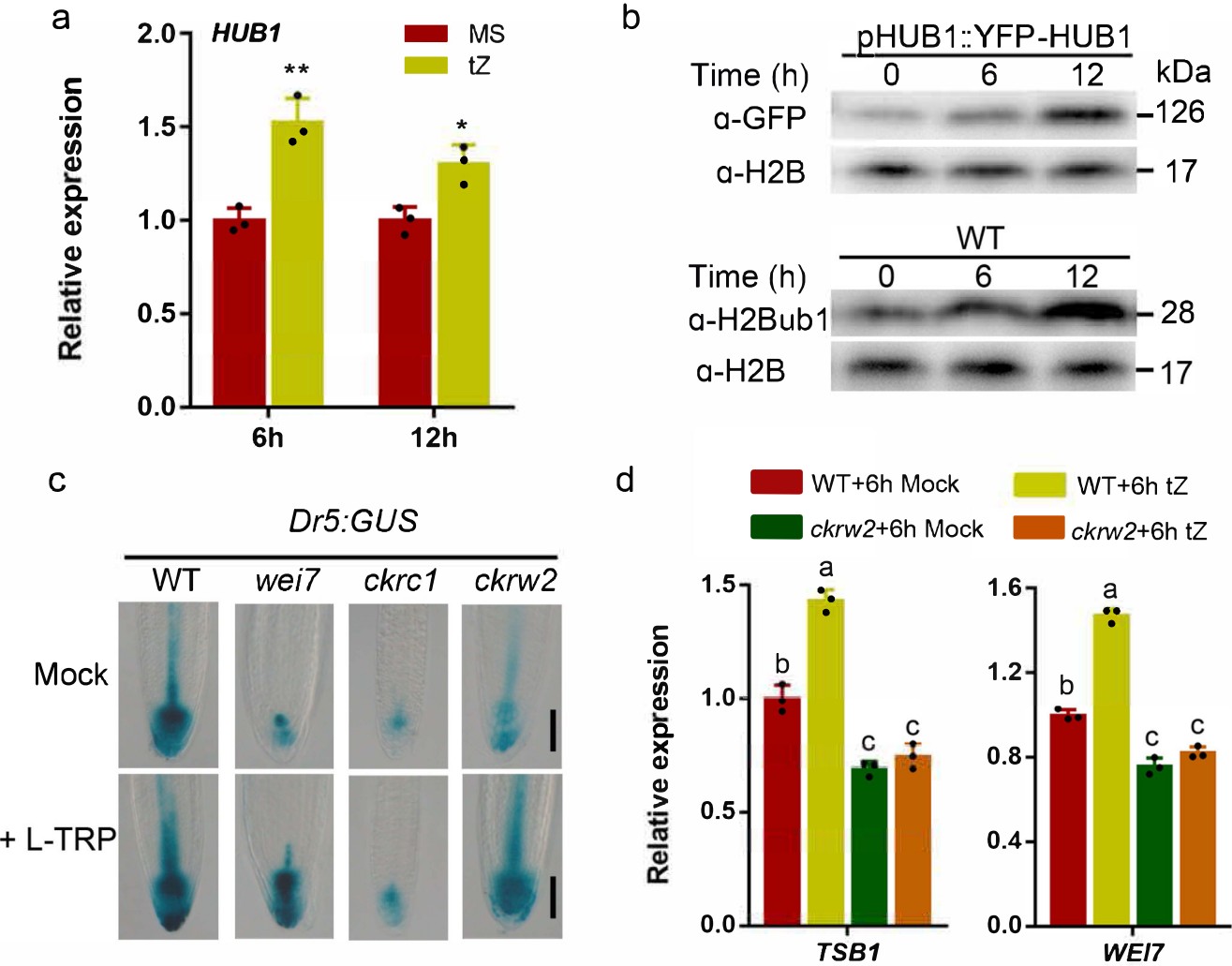

**Fig. 4 Evidences supporting the role of HUB1/2-mediated H2Bub1 in mediating regulation on auxin biosynthesis by CK in Arabidopsis. a, b** Effects of 1 μM tZ on *CKRW2/HUB1* expression and on the protein levels of YFP-HUB1 and H2Bub1. H2B was used as a loading control. **c** *DR5:GUS* activity showing the rescuing effect of 3 μM L-TRP on *wei7* and *ckrw2*, but bot on *ckrc1*. Bar = 50 μm. **d** Comparison of relative expression levels of *TSB1* and *WEI7* genes between WT and *ckrw2* mutant after tZ treatment. Data are presented as mean ± SD, three independent experiments, the asterisk **b** indicates a significant difference based on Student's *t* test (**$P < 0.01$, $0.05 > $*$P > 0.01$), and the letters **d** indicate a significant difference at $P < 0.05$, according to ANOVA followed by Tukey's multiple comparison tests. *ACTIN8* was used as an internal control (**a, d**).

mechanism via H2Bub1 can be used by CK as an effective way to regulate auxin biosynthesis through up-regulating HUB1/2 expression.

## Methods

**Plant material and growth conditions.** The conditions of germination and growth, as previously described[48], at 25 °C with a 16 h light/8 h dark photoperiod. For growth analyses, seedlings were grown on vertical MS (Murashige and Skoog) plates with 1.1% w/v agar supplemented with 10 g/L sucrose for 7 days.

Arabidopsis accession Col was used as WT. The *hub1-5* (SALK_044415; *AT2G44950*), *hub2-2* (SALK_071289; *AT1G55250*), and *hub1-5 hub2-2* mutants were kindly provided by Prof. Ligen Ma (Capital Normal University) and Dr. Baohong Zou (Nanjing Agricultural University), and *ami1* (N660737, *AT1G08980*) by Yan Guo (China Agriculture University). The *Dr5: GUS/ Dr5: GFP/DII-VENUS* marker line[49,50], *tsb1* (N8327; *AT5G54810*), *wei7* (*wei7 Dr5: GUS*) (N16436; *AT1G15220*), *yuc7* (N659416; *AT2G33230*) were purchased from the Nottingham Arabidopsis Stock Centre (NASC).

To generate *ckrw2/DR5: GUS* and *ckrw2 tsb1* mutants, the *ckrw2* mutant was crossed with *DR5: GUS* and *tsb1* mutant, respectively, and double homozygous mutants were obtained from the F2 generation.

**Phenotype characterization.** The degree of root curling/waving (DC) was calculated by dividing the distance between the two ends of seedlings' roots ($L_0$) by the length of their roots (L)[48]. Primary root length was measured after grown vertically on MS plates for 7 days[18,48].

For biochemical complementation experiments, seedlings were grown on a medium containing 0.01 μM tZ with auxin (0.01 μM 2, 4-D), and phenotypic observation and statistics were performed after 7 days of vertical cultivation.

For the L-TRP experiment, the 7-day-old seedlings were transferred to MS plates with or without 0.25 mM L-TRP and cultured for 2 weeks, and then the phenotype was observed[67].

**Gene cloning.** For map-based cloning, *ckrw2* plants from the F2 population of a cross between the *ckrw2* and Ler and Simple sequence length polymorphism (SSLP) markers (Supplementary Data 2) were used to locate the chromosomal position of *ckrw2* mutation, and the mutated gene was identified by WGRS (Hangzhou Guhe Information and Technology Co., Ltd., China. http://www. guheinfo.com/).

**Molecular complementation and GUS transgenic plants.** A 2 kb promoter sequence (2000 bp upstream of ATG) was amplified using pCKRW2-GUS-F (BamHI)/pCKRW2-GUS-R (NcoI) primers (Supplementary Data 2) and subcloned into a modified pCAMBIA1301 binary vector harboring a GUS gene to generate a pCKRW2:GUS reporter gene construct. For molecular complementation, the native *CKRW2* promoter (2000 bp upstream of ATG, Supplementary Data 2) and CDS were amplified by PCR, and placed in a pCAMBIA1300 vector. All amplified DNA fragments were verified by sequencing, then transformed into WT plants (for pCKRW2:GUS) or *ckrw2* mutant plants (for pCKRW2:CKRW2) by the floral dip

method using *Agrobacterium tumefaciens* (GV3101)[68]. The seeds of the transformants were stratified in 4 °C for 3 days, sterilized with 0.1% mercuric chloride, washed with sterilized water, and then isolated on MS medium containing 25 mg/L hygromycin B. The seedlings were transferred to the soil until maturity.

**Microscopic analysis**. For the GUS staining assay, 7-day-old seedlings were placed in the centrifugal tube, fixed with pre-cooled acetone for 20 min, washed twice with GUS base solution (50 mM $NaH_2PO_4·2H_2O$, 50 mM $Na_2HPO_4·2H_2O$, 10 mM EDTA·2Na, 0.1% Triton100, 0.5 mM $K_3[Fe(CN)_6]$, 0.5 mM $K_4Fe(CN)_6·3H_2O$), and incubated at 37 °C with 1 mM X-gluc (5-bromo-4-chloro-3-indolyl-β-D-glucuronide acid), and visualized under a microscope (Axio Imager.Z2, Zeiss, Germany).

For confocal microscopic analyses, 7-day-old seedlings were treated in propidium iodide (PI) solution (10 μg/mL) for 5 min (time can be adjusted according to the pre-experiment), then washed three times with ddH$_2$O, and visualized at 600–640 nm for PI and 500–560 nm for green fluorescent protein (GFP)/VENUS on a confocal microscope (TCS SP8, Leica, Germany). The *DR5: GUS /DR5:GFP/DII-VENUS* signal intensity of the root tip containing the GUS/ GFP/VENUS signal (approximately the first 200 μm from the root tip) was quantified by measuring the mean gray value with ImageJ[69].

For detecting the effect of CK on *HUB1*, seedlings were grown on MS medium for 7 days and then transferred to a liquid medium containing 1 μM tZ for 6 h. And then GUS staining and fluorescence observation were performed.

**RNA extraction and quantitative real-time PCR**. RNA was isolated using Trizol (No. B511321, Sangon Biotech) and reverse-transcribed using a reverse transcription kit (RR047, TAKARA). Quantitative RT-PCR was performed in a Real-time System (Bio-RAD CFX96, America) using TB Green (RR820A, Takara), with primers listed in Supplementary Data 2. The auxin synthesis gene expression analysis was carried out using the primary roots of the seedling grown on the MS medium for 7 days.

**Protein extraction and immunoblot analysis**. In order to detect the protein levels of HUB1 and H2Bub1, 7-day-old seedlings of pHUB1::YFP-HUB1 and WT were treated with tZ for a different time, respectively. For protein extraction and immunoblot analysis, a previously used experimental procedure was followed[70]. H2B was used as a loading control. The immunoblot analysis was carried out using an anti-H2B antibody (ab1790, Abcam) at a concentration of 0.1 μg/mL, anti-H2Bub1 antibody (MM-0029, Medimabs) at a concentration of 3–5 μg per sample, and an anti-GFP antibody (M20004, Abmart) at a concentration of 0.2 μg/mL. The signal was detected by a chemiluminescent horseradish peroxidase substrate system (No. C500044, Sangon).

**Chromatin immunoprecipitation (ChIP) assays**. ChIP assays were performed as previously described[71] using 7-day-old seedlings, which were grown on MS medium. In brief, the seedlings were vacuum cross-linked in 1% formaldehyde for 10 min, then 0.125 M glycine was added to the vacuum for 5 min to stop the cross-linking. To obtain 200–1000 bp DNA fragments, sonicate chromatin solution 5 times (5 s on, 15 s off in each time) by 50% power. Chromatin was immunoprecipitated using a specific anti-H2Bub1 antibody (MM-0029, Medimabs) and then specific protein A-agarose (11418033001, Roche). After the IP complex was pulled down and washed, the DNA was reverse cross-linked and then extracted using the phenol/chloroform method. The ChIP experiment used an equal amount of sample and protein A-agarose without antibody as a control. The ChIP DNA was finally analyzed by qPCR with three independent biological replicates.

**Statistics and reproducibility**. All results are expressed as the means ± standard deviation. The numbers of samples and replicates of experiments were shown as mentioned in the figure legends. Comparisons between groups were determined using Student's *t* test (significant difference at $0.01 < {}^*P < 0.05$, $^{**}P < 0.01$, $^{***}P < 0.001$) or ANOVA followed by Tukey's multiple comparison test (significant difference at $P < 0.05$). All data were analyzed using GraphPad Prism 7 software.

**Reporting summary**. Further information on research design is available in the Nature Research Reporting Summary linked to this article.

## Data availability

The nucleotide sequence of *CKRW2* was submitted to GenBank, and the accession number is BankIt2414347 ckrw2 MW431056. All other source data are included in the article as supplementary data 1–2. Uncropped scans of Western blots are shown in Supplementary Information. The unique biological materials of *ckrw2, ckrw2/Dr5:GUS, ckrw2/DII-VENUS* are available upon request to our lab.

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

## Acknowledgements

We are grateful to Professor Ligen Ma (Capital Normal University), Professor Yan Guo (China Agricultural University), and Dr. Baohong Zou (Nanjing Agricultural University) for providing seeds. We thank Mieke Van Lijsebettens (Ghent University, Belgium) for providing microarray data analysis for genes involved in auxin biosynthetic pathways. And, we thank the Core Facility of the School of Life Sciences, Lanzhou University. This study was supported by the National Natural Science Foundation of China (NSFC) (grant Nos. 31970713 and 31671458).

## Author contributions

G.Q.G. conceived and directed the research; L.Z. performed the major part of the research; P.L. isolated the *ckrw2* mutant; J.B., L.W., D.W.D., H.Q.L., J.J.L., Y.L.L., A.J.K. performed the experiments; W.L. assisted G.Q.G. to organize and coordinate the project; L.Z. wrote the draft of the paper and the final revision was accomplished by G.Q.G.; C.M.Z. and G.Q.G supervised L.Z. in her post-doctoral program. All authors discussed the data and the article.

## Competing interests

The authors declare no competing interests.
