## [Peer Review File · Communications Biology]

Reviewers' comments:

Reviewer #1 (Remarks to the Author):

The manuscript describes results from a mutant screen aiming at identification of factors from auxin biosynthetic pathway(s). The screen was based on auxin deficient phenotypes on cytokinin and resulted in a set of mutants with pleiotropic low auxin phenotypes that were rescued by auxin. The *ckrw2* mutant was identified by whole genome sequencing as an amino acid change causing an early terminator on Histone monoubiquitination1 gene. Allelic analysis and phenotypic rescue by complementation were used to confirm the gene identity. The gene *CKRW2* was shown to affect auxin biosynthetic pathway from TRP to IPA, and the mutant to have reduced HistoneH2BUB1 levels and reduced *WEI7* and *TSB1* gene expression levels. The work contributes in its part in resolving the auxin biosynthetic pathways, on the role of both histone modifications and CK in regulation of auxin homeostasis. The results are novel since they connect chromatin regulation with plant hormonal regulation.

Reviewer #2 (Remarks to the Author):

The authors performed map-based cloning and identified their previously isolated auxin-deficient mutant *ckrw2* as a loss-of-function mutant of *HUB1*. They showed indeed that the other *hub1/2* mutants exhibit root growth defects as does *ckrw2*. They further demonstrated that several genes involved in auxin synthesis are mis-expressed and their chromatin contains less H2Bub. Finally, they showed that *HUB1* expression is induced by cytokinin.

It is a good surprise that the authors found that *ckrw2* does not correspond to the T-DNA insertion mutation site. Mapping *ckrw2* as a *hub1* mutant allele does help to clarify the molecular basis of this previously reported *ckrw2*. Nevertheless, the *ckrw2* mutant (including its root growth defects) had been well described in the previous publication (Wu et al., 2015; Scientific Reports 5:11923). Also, *hub1/2* mutants had been very well described in literatures, including their growth and developmental defects, their responses to biotic and abiotic stimuli, their perturbed gene and genome expression, their impaired H2Bub, etc. (some literatures were cited in this current manuscript, but also many others were not cited). Thus, the content of this current manuscript has very limited novelty.

The authors tend to conclude that the expression of *HUB1* induced by cytokinin makes a link to their original identification of *ckrw2* in forward genetic screen for auxin-deficient mutants by cytokinin. However, there is little experimental data and the illustrated model (Fig 4e) does not have a bona fide content or is too speculative.

There is drastic lack of information on experimental method and data

treatment. Some additional specific comments:

- 1) Mutant screening was performed in the previous study. The first sentence in Methods is misleading.
- 2) Quantification of GUS blue-staining (Fig 1c, Extended Data Fig 5d) is unclear in several aspects. How intensity is measured (by which method, which type of equipment)? To which area of root? On how many individual roots? Roots of which age of plants?
- 3) Similarly, GFP/VENUS fluorescence quantification needs to be described.
- 4) For gene expression data (Fig 3a, 4b, 4d, Extended Data Fig 3), any internal control is used?
- 5) Data of Western blot (Extended Data Fig 4) showed more presence of H2Bub in *ckrw2* than in *hub1-5*, *hub2-2*. Is it possible that *ckrw2* corresponds to a partial loss-of-function mutant?

6) What means 'CHIP signal relative' in Fig 3c? relative to what? It is necessary to check H2B level in addition to H2Bub. Controls are required.

Reviewer #3 (Remarks to the Author):

In this manuscript, the authors have cloned the ckrw2 mutant and demonstrated that CKRW2 was an E3 ligase required for histone H2B monoubiquitination (H2Bub1). They further showed that without CKRW2, expression of several auxin biosynthesis related genes such as TRP2/TSB1, WEI7/ASB1, YUC7 and AMI1 was down-regulated. This work is interesting. The amount of work is substantial and the experiments appeared to be carefully completed. The conclusions are justified. My comments are minor and are related to the presentation.

Figure 2, two independent complementation lines should be shown to rule out off-target or other weird effects.

The authors may not know the latest auxin literature. For example, Amidase I is likely not important for auxin biosynthesis "J Genet Genomics" 2020 Mar 20;47(3):157-165. doi: 10.1016/j.jgg.2020.02.009.

Line 62, a terminator (TGG > TAG) , Replace "terminator" with "stop codon"

Line 89, " To know how ckrw2 mutation affected auxin homeostasis", replace "know" with "investigate"

Reviewer #1 (Remarks to the Author):

The manuscript describes results from a mutant screen aiming at identification of factors from auxin biosynthetic pathway(s). The screen was based on auxin deficient phenotypes on cytokinin and resulted in a set of mutants with pleiotropic low auxin phenotypes that were rescued by auxin. The *ckrw2* mutant was identified by whole genome sequencing as an amino acid change causing an early terminator on Histone monoubiquitination1 gene. Allelic analysis and phenotypic rescue by complementation were used to confirm the gene identity. The gene *CKRW2* was shown to affect auxin biosynthetic pathway from TRP to IPA, and the mutant to have reduced HistoneH2Bub1 levels and reduced *WEI7* and *TSB1* gene expression levels. The work contributes in its part in resolving the auxin biosynthetic pathways, on the role of both histone modifications and CK in regulation of auxin homeostasis. The results are novel since they connect chromatin regulation with plant hormonal regulation.

Reply: Many thanks for your recognition!

Reviewer #2 (Remarks to the Author):

It is a good surprise that the authors found that *ckrw2* does not correspond to the T-DNA insertion mutation site. Mapping *ckrw2* as a *hub1* mutant allele does help to clarify the molecular basis of this previously reported *ckrw2*. Nevertheless, the *ckrw2* mutant (including its root growth defects) had been well described in the previous publication (Wu et al., 2015; Scientific Reports 5:11923). Also, *hub1/2* mutants had been very well described in literatures, including their growth and developmental defects, their responses to biotic and abiotic stimuli, their perturbed gene and genome expression, their impaired H2Bub, etc. (some literatures were cited in this current manuscript, but also many others were not cited). Thus, the content of this current manuscript has very limited novelty.

Reply: Thank you for your suggestion, we have carefully checked the manuscript and cited some literatures (Dhawan et al., 2009; Feng et al., 2018; Ma et al., 2019; Zhang et al., 2020; Zhang et al., 2015; Zhao et al., 2020).

The authors tend to conclude that the expression of HUB1 induced by cytokinin makes a link to their original identification of *ckrw2* in forward genetic screen for auxin-deficient mutants by cytokinin. However, there is little experimental data and the illustrated model (Fig 4e) does not have a bona fide content or is too speculative.

Reply: Here may be a misunderstanding: by saying “how the regulation of auxin biosynthesis by CK was related to H2Bub1” did not mean we want to conclude that the expression of HUB1 induced by cytokinin makes a link to our original identification of *ckrw2* in forward genetic screen for auxin-deficient mutants by cytokinin, which is simply due to the endogenous auxin-deficiency, as reported in our

previous reports, and *ckrw2* was only one of the auxin-deficient mutants screened by CK(Wu et al., 2015).

We modified the model shown in figure 4e to show that besides HUB1/2 transcription-H2Bub1-WEI7/TSB1, CK also regulates auxin biosynthesis through the H2Bub-independent transcription of TAA1 and YUCs, as reported in our cited refs in the manuscript.

There is drastic lack of information on experimental method and data treatment.

Reply: The required information on experimental method and data treatment were provided in this revised manuscript, please see lines 288 to 293, lines 316 to 332, and lines 341 to 349 in the text.

Some additional specific comments:

1) Mutant screening was performed in the previous study. The first sentence in Methods is misleading.

Reply: Thank you for your reminding. We have deleted the first sentence in Methods because it has been described clearly in the results section.

2) Quantification of GUS blue-staining (Fig 1c, Extended Data Fig 5d) is unclear in several aspects. How intensity is measured (by which method, which type of equipment)? To which area of root? On how many individual roots? Roots of which age of plants?

3) Similarly, GFP/VENUS fluorescence quantification needs to be described.

Reply: We used seedlings grown on MS medium for 7 days for GUS staining or GFP/VENUS fluorescence observation. The intensity of GFP/VENUS fluorescence or GUS staining (approximately to the first 200 μm from the root tip) was quantified by measuring the mean gray value with ImageJ (Figure 1 in Response to the Reviewer 2). At least 20-35 roots in each lines are stained each time, three biological replicates.

For better understanding, we changed the "bar chart" to "boxplots". We have revised Fig. 1 and the legend (as shown below Fig.1), please see lines 52 to 56 in the text.

Figure 1 in Response to the Reviewer 2.

Fig.1 | Auxin-deficient phenotypes in *ckrw2* mutant, showing its waving roots after 7 days of vertical culture on MS with 0.01 1iM tZ, which can be rescued by 0.01 1iM 2, 4-D. Bar = 5 mm (a). b, Results of statistical analysis on root curling/waving degrees (DC). Data are presented as mean \pm SD (n=30-35), three biologically independent experiments, the letters indicate a significant difference at $P < 0.05$ according to ANOVA followed by Tukey's multiple comparison tests (b). c-h, Auxin activity in root tips revealed by the expressions of *Dr5: GUS/GFP/DII-VENUS*, respectively. The seedlings grown on MS medium for 7 days were used for GUS staining and fluorescence intensity observation (approximately to the first 200 μ M from the root tip). Each circle represents the measurement from an individual root. Boxplots span the first to the third quartiles of the data. Whiskers represent minimum and maximum values. A line in the box represents the mean. "n" represents the number of roots used in this experiment. Student's t-test, was used for statistical analyses. *** $P < 0.001$. Bars=50 μ m in (c, e, g).

4) For gene expression data (Fig 3a, 4b, 4d, Extended Data Fig 3), any internal control is used?

Reply: Yes. *ACTIN8* was used as an internal control, as indicated in the revised figure legends.

5) Data of Western blot (Extended Data Fig 4) showed more presence of H2Bub in *ckrw2* than in *hub1-5*, *hub2-2*. Is it possible that *ckrw2* corresponds to a partial loss-of-function mutant?

Reply: It is unlikely that *ckrw2* is a partial loss-of-function mutant, since this mutation caused a very early stop in HUB1 protein translation (Figure 2 in Response to the Reviewer 2). It appears that the noted slight difference in the weak bands intensity are not significantly enough to draw such a conclusion.

Figure 2 in Response to the Reviewer 2. (A) Gene structures and characterization of the steady state mRNA levels of HUB1 and HUB2 by RT-PCR in the *hub1* and *hub2* mutants. The primers used to detect the transcripts are indicated as P1 to P8. Pairs of P3/P4 and P7/P8 were used to detect the full-length transcripts of HUB1 and HUB2, respectively; Pairs of P1/P2 and P5/P6 were used to detect the partial transcripts of HUB1 and HUB2, respectively. (B) Gene structure of AT2g44950 (*HUB1*). Exons are represented by filled black boxes, introns by lines, untranslated regions by filled gray boxes, and T-DNA insertions by triangles.

6) What means 'CHIP signal relative' in Fig 3c? relative to what? It is necessary to check H2B level in addition to H2Bub. Controls are required.

Reply: In Fig 3c, CHIP signal relative to input. Each ChIP value was normalized to its respective input DNA value (defined as 100%)(Menard et al., 2014). *LCRa* and *FLCP4* were used as negative and positive control, respectively (shown in Extended Data Fig. 4), as reported in other previous studies (Cao et al., 2008; Menard et al., 2014).

As you suggested, we checked H2B level in WT and *ckrw2* mutant, finding no significant difference between and the WT and *ckrw2* mutant, as shown below in Figure 3 B in Response to the Reviewer 2, consistent with other reports(Cao et al., 2008) (Figure 3 in Response to the Reviewer 2).

Figure 3 in Response to the Reviewer 2. (A) Detection of H2Bub1 in wildtype and mutant plants. The upper band was monoubiquitinated Flag H2B, and the lower band was Flag-H2B. The bands were detected by anti-Flag antibody. H2Bub1, monoubiquitinated H2B; MW, molecular weight. (B) The level of H2B protein in WT and *ckrw2*. CBB, Coomassie brilliant blue-stained RbcS protein, loading control. α -H2B, anti-Histone H2B antibody.

Reviewer #3 (Remarks to the Author):

Figure 2, two independent complementation lines should be shown to rule out off-target or other weird effects.

Reply: Thank you for your suggestions. Phenotypic rescue of *ckrw2* mutant also occurred in other complementation lines (CKRW2-L4), please see Fig. 2 b in the text.

The authors may not know the latest auxin literature. For example, Amidase I is likely not important for auxin biosynthesis “J Genet Genomics” 2020 Mar 20;47(3):157-165. doi: 10.1016/j.jgg.2020.02.009.

Reply: Thanks. This paper is very new and has been cited in our revised version. Our results show that under normal conditions, *ami1* has the same phenotype as the mutants in this cited paper, and there is no obvious developmental defects. In our manuscript, HUB1 not only regulates *AM11*, but also regulates other important genes for auxin synthesis. Our conclusion is not contradictory to the conclusion of this paper.

Line 62, a terminator (TGG > TAG), Replace “terminator” with “stop codon”
 Line 89, “To know how *ckrw2* mutation affected auxin homeostasis”, replace “know” with “investigate”

Reply: We have made correction according to your comments. Thank you very much.

Thank you very much for your comments and suggestions!

Reference

Cao, Y., Dai, Y., Cui, S., and Ma, L. (2008). Histone H2B monoubiquitination in the chromatin of FLOWERING LOCUS C regulates flowering time in Arabidopsis. *The Plant cell* 20, 2586-2602.

Dhawan, R., Luo, H., Foerster, A.M., Abuqamar, S., Du, H.N., Briggs, S.D., Mittelsten Scheid, O., and Mengiste, T. (2009). HISTONE MONOUBIQUITINATION1 interacts with a subunit of the mediator complex and regulates defense against necrotrophic fungal pathogens in Arabidopsis. *The Plant cell* 21, 1000-1019.

Feng, H., Li, X., Chen, H., Deng, J., Zhang, C., Liu, J., Wang, T., Zhang, X., and Dong, J. (2018). GhHUB2, a ubiquitin ligase, is involved in cotton fiber development via the ubiquitin-26S proteasome pathway. *Journal of experimental botany* 69, 5059-5075.

Ma, S., Tang, N., Li, X., Xie, Y., Xiang, D., Fu, J., Shen, J., Yang, J., Tu, H., Li, X., *et al.* (2019). Reversible Histone H2B Monoubiquitination Fine-Tunes Abscisic Acid Signaling and Drought Response in Rice. *Molecular plant* 12, 263-277.

Menard, R., Verdier, G., Ors, M., Erhardt, M., Beisson, F., and Shen, W.H. (2014). Histone H2B monoubiquitination is involved in the regulation of cutin and wax composition in Arabidopsis thaliana. *Plant & cell physiology* 55, 455-466.

Zhang, B., Sztojka, B., Seyfferth, C., Escamez, S., Miskolczi, P., Chantreau, M., Bako, L., Delhomme, N., Gorzsas, A., Bhalerao, R.P., *et al.* (2020). The chromatin-modifying protein HUB2 is involved in the regulation of lignin composition in xylem vessels. *Journal of experimental botany* 71, 5484-5494.

Zhang, Y., Li, D., Zhang, H., Hong, Y., Huang, L., Liu, S., Li, X., Ouyang, Z., and Song, F. (2015). Tomato histone H2B monoubiquitination enzymes SIHUB1 and SIHUB2 contribute to disease resistance against Botrytis cinerea through modulating the balance between SA- and JA/ET-mediated signaling pathways. *BMC plant biology* 15, 252.

Zhao, J., Chen, Q., Zhou, S., Sun, Y., Li, X., and Li, Y. (2020). H2Bub1 Regulates RbohD-Dependent Hydrogen Peroxide Signal Pathway in the Defense Responses to Verticillium dahliae Toxins. *Plant physiology* 182, 640-657.

Wu, L., Luo, P., Di, D.W., Wang, L., Wang, M., Lu, C.K., Wei, S.D., Zhang, L., Zhang, T.Z., Amakorova, P., *et al.* (2015). Forward genetic screen for auxin-deficient mutants by cytokinin. *Scientific reports* 5, 11923.

REVIEWERS' COMMENTS:

Reviewer #2 (Remarks to the Author):

The authors responded to most of my points raised in the previous version. Nevertheless, there are still some issues that need to be taken into consideration.

The authors described in revision that 'The DR5:GUS/DR5:GFP/DII-VENUS signal intensity of the root tip containing the GUS/GFP/VENUS signal (approximately to the first 200 μm from the root tip) was quantified by measuring the mean gray value with ImageJ'. It is not clear why GFP fluorescence has not been measured directly. A gray value might not precisely reflect expression level. GUS and GFP reporters provide nice information for tissue-specific expression pattern, but quantitative analysis of expression levels should be better done by qRT-PCR.

I am not satisfied by the authors' response regarding the illustrative model (Fig 4e). It is known in general that epigenetic marks (e.g. H2Bub) play co-factor but not initiator roles in transcription regulation. However, the model and the last sentence of the main text 'By up-regulating HUB1/2 expression, CK can regulate auxin homeostasis to participate in the control of plant growth and development' clearly points to a new dimension of H2Bub function in auxin synthesis gene transcription. It is not convincing that an up-regulation of roughly 1.5-fold of HUB1/2 expression by CK (Fig 4d) could place H2Bub in such a dimension role. Also, the model is not described for reader understanding. UBC1 and UBC2 have redundant roles and acts as E2 Ub-conjugating enzymes (Gu et al., Plant Journal; Xu et al. Plant Journal).

Language editing needs further improvement.

Response to the reviewers' comments

Reviewer #2 (Remarks to the Author)

The authors responded to most of my points raised in the previous version.

Nevertheless, there are still some issues that need to be taken into consideration.

The authors described in revision that 'The DR5:GUS/DR5:GFP/DII-VENUS signal intensity of the root tip containing the GUS/GFP/VENUS signal (approximately to the first 200 μm from the root tip) was quantified by measuring the mean gray value with ImageJ'. It is not clear why GFP fluorescence has not been measured directly. A gray value might not precisely reflect expression level. GUS and GFP reporters provide nice information for tissue-specific expression pattern, but quantitative analysis of expression levels should be better done by qRT-PCR.

Reply: We do not think that qRT-PCR analysis of expression levels is better than directly measuring the visual signal of those sensitive auxin abundance/activity markers to evaluate auxin level in root tips, because the expression of DII-VENUS is under the control of the constitutive 35S promoter. Regarding the analysis of color/gray value by ImageJ, we refer to (Zhang et al., 2019). Since DII-VENUS is more directly related to auxin abundance than DR5-GFP/GUS, we removed the picture of DR5-GFP/GUS results into the supplemental data in the present revised manuscript.

I am not satisfied by the authors' response regarding the illustrative model (Fig 4e). It is known in general that epigenetic marks (e.g. H2Bub) play co-factor but not initiator roles in transcription regulation. However, the model and the last sentence of the main text 'By up-regulating HUB1/2 expression, CK can regulate auxin homeostasis to participate in the control of plant growth and development' clearly points to a new dimension of H2Bub function in auxin synthesis gene transcription. It is not convincing that an up-regulation of roughly 1.5-fold of HUB1/2 expression by CK (Fig 4d) could place H2Bub in such a dimension role. Also, the model is not described for reader understanding. UBC1 and UBC2 have redundant roles and acts as E2

Ub-conjugating enzymes (Gu et al., Plant Journal; Xu et al. Plant Journal).

Reply: We did not want in the model to express the idea, as this reviewer indicated, that H2Bub1 play initiator but not co-factor roles in transcription regulation. It seems that the model in Fig 4e is not much helpful to understand the role of H2Bub1 in regulating auxin biosynthesis by CK, therefore we deleted this figure and rewrite the sentences in the last paragraph.

Language editing needs further improvement.

Reply: In the process of revision, we have made some necessary language editions.

Reference

Zhang, Y., He, P., Ma, X., Yang, Z., Pang, C., Yu, J., Wang, G., Friml, J., and Xiao, G. (2019). Auxin-mediated statolith production for root gravitropism. *The New phytologist* 224, 761-774.

Response to the reviewers' comments

Reviewer #2 (Remarks to the Author)

The authors responded to most of my points raised in the previous version. Nevertheless, there are still some issues that need to be taken into consideration.

The authors described in revision that 'The DR5:GUS/DR5:GFP/DII-VENUS signal intensity of the root tip containing the GUS/GFP/VENUS signal (approximately to the first 200 μ m from the root tip) was quantified by measuring the mean gray value with ImageJ'. It is not clear why GFP fluorescence has not been measured directly. A gray value might not precisely reflect expression level. GUS and GFP reporters provide nice information for tissue-specific expression pattern, but quantitative analysis of expression levels should be better done by qRT-PCR.

Reply: We do not think that qRT-PCR analysis of expression levels is better than directly measuring the visual signal of those sensitive auxin abundance/activity markers to evaluate auxin level in root tips, because the expression of *DII-VENUS* is under the control of the constitutive 35S promoter. Regarding the analysis of color/gray value by ImageJ, we refer to (Zhang et al., 2019). Since *DII-VENUS* is more directly related to auxin abundance than *DR5-GFP/GUS*, we removed the picture of *DR5-GFP/GUS* results into the supplemental data in the present revised manuscript.

I am not satisfied by the authors' response regarding the illustrative model (Fig 4e). It is known in general that epigenetic marks (e.g. H2Bub) play co-factor but not initiator roles in transcription regulation. However, the model and the last sentence of the main text 'By up-regulating HUB1/2 expression, CK can regulate auxin homeostasis to participate in the control of plant growth and development' clearly points to a new dimension of H2Bub function in auxin synthesis gene transcription. It is not convincing that an up-regulation of roughly 1.5-fold of HUB1/2 expression by CK (Fig 4d) could place H2Bub in such a dimension role. Also, the model is not described for reader understanding. UBC1 and UBC2 have redundant roles and acts as E2 Ub-conjugating enzymes (Gu et al., *Plant Journal*; Xu et al. *Plant Journal*).

Reply: We did not want in the model to express the idea, as this reviewer indicated, that H2Bub1 play initiator but not co-factor roles in transcription regulation. It seems that the model in Fig 4e is not much helpful to understand the role of H2Bub1 in regulating auxin biosynthesis by CK, therefore we deleted this figure and rewrite the sentences in the last paragraph.

Language editing needs further improvement.

Reply: In the process of revision, we have made some necessary language editions.

Reference

Zhang, Y., He, P., Ma, X., Yang, Z., Pang, C., Yu, J., Wang, G., Friml, J., and Xiao, G. (2019). Auxin-mediated statolith production for root gravitropism. *The New phytologist* **224**, 761-774.